# Photodynamic Effects with 5-Aminolevulinic Acid on Cytokines and Exosomes in Human Peripheral Blood Mononuclear Cells

**DOI:** 10.3390/biomedicines10020232

**Published:** 2022-01-21

**Authors:** Kristian Espeland, Andrius Kleinauskas, Petras Juzenas, Andreas Brech, Sagar Darvekar, Vlada Vasovic, Trond Warloe, Eidi Christensen, Jørgen Jahnsen, Qian Peng

**Affiliations:** 1Department of Gastroenterology, Akershus University Hospital, N-1478 Lorenskog, Norway; jorgen.jahnsen@medisin.uio.no; 2Department of Pathology, Norwegian Radium Hospital, Oslo University Hospital, N-0310 Oslo, Norway; andrius.kleinauskas@rr-research.no (A.K.); petras.juzenas@rr-research.no (P.J.); sagar.darvekar@rr-research.no (S.D.); Vlada.Vasovic@rr-research.no (V.V.); trond.warloe@gmail.com (T.W.); eidi.christensen@ntnu.no (E.C.); 3Institute of Clinical of Medicine, Faculty of Medicine, University of Oslo, N-0372 Oslo, Norway; abrech@rr-research.no; 4Department of Molecular Cell Biology, Institute for Cancer Research, Norwegian Radium Hospital, Oslo University Hospital, N-0372 Oslo, Norway; 5Department of Clinical and Molecular Medicine, Norwegian University of Science and Technology, N-7030 Trondheim, Norway; 6Department of Dermatology, St. Olavs Hospital, Trondheim University Hospital, N-7030 Trondheim, Norway; 7Department of Optical Science and Engineering, School of Information Science and Technology, Fudan University, Shanghai 200433, China

**Keywords:** 5-aminolevulinic acid (ALA), photodynamic therapy (PDT), photodynamic diagnosis, protoporphyrin IX (PpIX), peripheral blood mononuclear cells (PBMCs), cytokine, exosome, electron microscopy, flow cytometry

## Abstract

Photodynamic therapy (PDT) with 5-aminolevulinic acid (ALA), a precursor to the potent photosensitizer, protoporphyrin IX (PpIX), is an established modality for several malignant and premalignant diseases. This treatment is based on the light-activated PpIX in targeted lesions. Although numerous studies have confirmed the necrosis and apoptosis involved in the mechanism of action of this modality, little information is available for the change of exosome levels after treatment. We report from the first study on the effects of ALA-PDT on cytokines and exosomes of human healthy peripheral blood mononuclear cells (PBMCs). The treatment reduced the cytokines and exosomes studied, although there was variation among individual PBMC samples. This reduction is consistent with PDT-mediated survivals of subsets of PBMCs. More specifically, the ALA-PDT treatment apparently decreased all pro-inflammatory cytokines included, suggesting that this treatment may provide a strong anti-inflammatory effect. In addition, the treatment has decreased the levels of different types of exosomes, the HLA-DRDPDQ exosome in particular, which plays an important role in the rejection of organ transplantation as well as autoimmune diseases. These results may suggest future therapeutic strategies of ALA-PDT.

## 1. Introduction

Photodynamic therapy (PDT), an established treatment modality for several malignant and pre-malignant diseases, uses a combination of a lesion-localizing photosensitizing agent with light irradiation to induce photochemical and photobiological reactions in the presence of oxygen. These reactions lead to irreversible photodamage to the lesion. During this dynamic process, the absorbed light energy by the photosensitizer can be transferred to molecular oxygen to generate reactive oxygen species (ROS), including singlet oxygen (^1^O_2_). These species react further with cellular components to cause apoptosis and/or necrosis [1,2].

PDT with chemically synthesized sensitizers, has a major side-effect of skin phototoxicity, limiting clinical PDT to a great extent. Thus, considerable interest has been directed towards developing a PDT regimen with an endogenous potent photosensitizer, protoporphyrin IX (PpIX). 5-aminolevulinic acid (ALA), a naturally occurring amino acid formed from glycine and succinyl CoA, is a precursor to PpIX in the heme biosynthesis pathway. By adding exogenous ALA, the naturally occurring PpIX may accumulate in cells [3,4]. This intracellular PpIX accumulation has been exploited for its application in photodiagnosis and PDT [5]. Clinically, systemic administration of ALA is used for the PpIX fluorescence-guided surgical resection of glioma [6]; while topically applied ALA-PDT has already been approved by United States Food and Drug Administration (FDA) and European Medicines Agency (EMA) for actinic keratosis and basal cell carcinoma of the skin [4].

Cytokines are a family of small proteins that are involved in growth and activities of blood cells including immune cells. They are largely produced by immune cells such as macrophages, T-cells, B-cells, and non-immune cells including endothelial cells and fibroblasts. Once released, they signal and modulate the immune system.

Exosomes, with a diameter of about 50–150 nm, are extracellular lipid bilayer vesicles with an endosomal origin of cells. They carry nucleic acids, proteins, lipids, and other bioactive molecules. Exosomes can be secreted by healthy and diseased cells through different pathways [7]. They can merge with cells and are involved in signaling between cells, but it is still not clear how exosomes communicate with cells and which roles they play in a biological system [8,9]. It is known that PDT can lead to release of exosomes from tumor cells [10,11]. Further, Zhao et al. have shown that ALA-PDT-derived exosomes specifically enhanced anti-tumor immunity in squamous cell carcinoma cells [12].

Recently, the use of ALA-PDT has extended to treat hematological malignant cells [13] and autoimmune disease [14]. However, the mechanism of action on killing diseased and normal cells in the blood is far from understood. In this report, we have studied the photodynamic effects on cytokines and exosomes of human normal peripheral blood mononuclear cells (PBMCs) with ALA.

## 2. Materials and Methods

### 2.1. Chemicals

5-Aminolevulinic acid (ALA) was obtained from Sigma Aldrich (St. Louis, MO, USA). A fresh stock solution of ALA was prepared in Phosphate Buffered Saline (PBS) (VWR Life Science, Solon, OH, USA) to a concentration of 1 M and kept at 4 °C. This was further diluted to reach a concentration of 3 mM for each experiment. All chemicals used were of the highest purity commercially available.

### 2.2. Isolation and Culture of PBMCs

The human buffy coat samples from 4 anonymous healthy donors (Regional Committee for Medical Research Ethics, REK-S 03280) were obtained from the Blood Bank, Oslo University Hospital, Norway. The isolation of PBMCs from the buffy coats was done using the Lymphoprep density gradient solution (Axis-Shield, Oslo, Norway) in SepMate™ (Stemcell Technologies, Cambridge, UK) 50 mL tubes. Fifteen mL of Lymphoprep were pipetted to the SepMate™ tube. The buffy coats were diluted with an equal volume of RPMI-1640 growth medium (Gibco, Grand Island, NY, USA) containing 2% fetal bovine serum (FBS) (Biological Industries Israel Beit-Haemek Ltd., Kibbutz Beit-Haemek, Israel) and layered carefully above the density gradient solution. The tubes were centrifugated at 1200× *g* for 10 min at room temperature with the brake on. The top layer was poured off for maximally 2 s to a new falcon tube and washed twice with the growth medium with 2% FBS, firstly at 300× *g* for 15 min and secondly at 300× *g* for 10 min. The isolated PBMCs were then gradually frosted using a CoolCell™ (Corning Incorporated, Corning, NY, USA) and stored at −80 °C before use.

PBMCs had been defrosted and incubated in RPMI-1640 growth medium with 10% exosome-depleted FBS(Gibco), 100 units/mL penicillin (Gibco), 10 µg/mL streptomycin (Gibco), and 2 mM L-glutamine(Gibco) at 37 °C in a humidified atmosphere with 5% CO_2_ for 24 h before experiments were performed.

### 2.3. Incubation with ALA and ALA Induced PpIX Production in PBMCs

The incubated PBMCs were changed with fresh medium and one mL of 2 × 10^6^ cells/mL was added per well in a 24-well plate. The cells were then incubated with ALA at the concentration of 3 mM for 4 h in the dark at 37 °C. This ALA dose was chosen because of a therapeutic effect of ALA-PDT on PBMCs in our previous study [15]. Intracellular ALA-induced PpIX in the PBMC subsets was measured with flow cytometry (as described in Section 2.6) using a 405 nm violet laser for the excitation and a 660/20 BP emission filter for the detection of PpIX geometric mean fluorescence intensity.

### 2.4. Light Source

The Light Emitting Diode (LED) lamp of PhotoCure Aktilite CL 128 (Galderma SA, Lausanne, Switzerland) was used at the wavelength of 630 nm with a fluence rate of 100 mW/cm^2^ for 30 min giving a total light dose of 180 J/cm^2^.

### 2.5. PDT Treatment of PBMCs

Four hours after ALA incubation the cells were irradiated by the LED lamp at room temperature. After light exposure, the cells continued to be incubated for another 48 h in the dark at 37 °C with the same ALA-containing medium to keep cytokines and exosomes. The cells incubated with ALA alone were also included as controls. The survivals of PBMC subsets were measured with flow cytometry as described in Section 2.6.

### 2.6. Flow Cytometry Analysis

After incubation with ALA alone or ALA plus light, the cells were centrifuged and the supernatants were kept at −80 °C for the isolation, labelling, and measurements of cytokines and exosomes as described in Section 2.7. The cells were then incubated in PBS (2% FBS) with an antibody concentration of 2 µL/mL and AnnexinV concentration of 50 µL/mL. The cells were washed with PBS (2% FBS), centrifuged, and the supernatants removed. The amounts of ALA-induced PpIX (1 h antibody incubation) and cell viabilities (1.5 h antibody incubation) in the individual subsets of PBMCs were measured with 4 antibody/dye combination methods: (1) CD3-FITC (Invitrogen, Carlbad, CA, USA), CD19-PE (ImmunoTools GmbH, Friesoythe, Germany), Fixable viability dye-eF450 (Invitrogen), AnnexinV-AF647 (Life Technologies, Eugene, OR, USA); (2) CD3-FITC, CD56-APC-eF780 (Invitrogen), Fixable viability dye-eF450, AnnexinV-PE (Invitrogen); (3) CD4-FITC (eBioscience, San Diego, CA, USA), CD8-PE (Invitrogen), Fixable viability dye-eF450, AnnexinV-AF647; (4) CD11c-FITC (ImmunoTools), CD14-PE/Cy7 (Invitrogen), Fixable viability dye-eF450, Annexin V-AF647. The measurements were done using a Cytoflex S cytometer (Beckman Coulter Life Sciences, Indianapolis, IN, USA) with the Cytexpert software (Version 2.1, Beckman Coulter); and the analyses were performed using FlowJo software (Version 10, Treestar, Ashland, OR, USA).

### 2.7. Isolation, Labelling, and Measurement of Cytokines and Exosomes

Isolation and labelling of cytokines and exosomes are based on the Exosome Isolation Kit CD63 (Magnetic Exosome Isolation Beads) and MACSPlex Exosome Kit (MACSPlex Exosome Capture Beads, MACSPlex Exosome Detection Reagent CD63 and 3 Buffer Solutions) (both kits from MACS Miltenyi Biotech GmbH, Bergisch Gladbach, Germany). On day 1, after thawing 900 µL of a clean supernatant sample (containing both cytokines and exosomes) were transferred to an Eppendorf tube (Eppendorf AG, Hamburg, Germany). Fifty µL of Magnetic Exosome Isolation Beads were added to the sample and the mixture was put on a shaker (Vortex T Genie-2, Scientific Industries, Bohemia, NY, USA) at Vortex speed 3 for 8 h at room temperature and stored overnight in a refrigerator at 4 °C before magnetic isolation of exosomes.

On day 2, a proposed µ-column (same manufacturer) was placed in a magnetic µMACS separator that was attached to a MACS multistand (same manufacturer). The µ-column was added with 100 µL of the Equilibration Buffer (same manufacturer) followed by washing with 100 µL of the Isolation Buffer for three times. The magnetically isolated sample from day 1 was then added to go through the µ-column for cytokine collection first (with no need of magnetic bead’s help). Subsequently, the µ-column was washed four times using 200 µL of the Isolation Buffer. After washing, the µ-column was moved away from the magnetic separator and immediately flushed into a clean Eppendorf tube with 100 µL of the Isolation Buffer by using a plunger (Figure 1). For labelling of the exosomes, 15 µL of the MACSPlex Exosome Capture Beads were added to the same sample and mixed carefully. A control sample without exosomes was also included using the same Capture Beads. The sample was then put on the shaker for 1 h and stored overnight in a refrigerator at 4 °C.

On day 3, the sample was put on the shaker at Vortex speed 3 for 5 h. One mL of the MACSPlex Buffer was added to the sample and centrifuged at 12,000 rpm for 10 min. The supernatant was removed and 5 µL of the MACSPlex Exosome Detection Reagent CD63 (for detection of surface markers of exosomes) was added before a 3 h incubation on the shaker (Figure 1). The sample was then washed twice by adding 1 mL of the MACSPlex Buffer, centrifuged at 12,000 rpm for 10 min, and the supernatant removed again.

Cytokines were labelled using Bio-plex Pro human Cytokine Grp 1 panel 27-Plex kit (Bio-Rad Laboratories, Inc., Hercules, CA, USA) (Figure 1). This was done by firstly mixing 50 µL of a sample with 50 µL of a bead solution in an Eppendorf tube. The bead solution was made by dilution of 2 µL beads in 50 µL assay buffer. After mixing, this was incubated for 30 min and washed by adding 1 mL of wash buffer (diluted 10 times in distilled water), centrifuged at 10,000 rpm for 5 min, and the supernatant removed. Then, 20 µL of detection antibody solution (2 µL detection antibody in 50 µL detection antibody diluent) was added and the sample was incubated for 30 min and washed as mentioned above before the supernatant was aspirated. Twenty µL of streptavidin—PE solution (1 µL streptavidin-PE in 100 µL detection antibody diluent) was then added and incubated for 10 min. The same wash step was repeated, and supernatant removed. Finally, 50 µL of the assay buffer were added per sample.

The cytokine and exosome samples were measured with the Cytoflex S cytometer as described in Section 2.6.

### 2.8. Electron Microscopy of Exosomes

Samples were deposited on glow-discharged formvar/carbon-coated mesh grids (100 mesh) and negatively stained with 2% uranyl acetate for 2 min. The observation of grids was done in a JEOL-JEM 1230 at 80 kV and images were acquired with a Morada camera (Olympus, Hamburg, Germany) using the iTem software.

### 2.9. Statistical Analyses

The results of the study are plotted and presented as maximum, minimum, and average values to see the spread.

## 3. Results

### 3.1. ALA-Induced PpIX Production in PBMCs

Figure 2 shows the PpIX production in the subpopulations of PBMCs from four different donors after ALA incubation at a concentration of 3 mM in the dark for 4 h.

Most subsets of PBMCs produced PpIX from ALA with average amounts of 200% to 300% of the control samples without ALA. However, CD8^+^ T cells produced little ALA-induced PpIX, while CD3^−^CD19^+^ B cells had a higher average amount of PpIX with a big variation in the samples from four different donors (Appendix A).

### 3.2. Dark Toxicity of PBMCs with ALA Alone

No apparent dark toxicities on various subpopulations of PBMCs were seen (Figure 3), although there are some variations among samples from donors. In the case of CD3^−^CD56^+^ of Natural Killer (NK) cells, however, there was a killing effect with ALA alone (Figure 3, Appendix A).

Generally, ALA alone at the concentration of 3 mM for a 4-h incubation does not cause any cytotoxicity. In this study, since the cells were incubated with ALA for 4 h followed by another 48 h to keep cytokines and exosomes, such a relatively long ALA incubation might result in some toxicity on certain subsets of PBMCs.

### 3.3. PDT of PBMCs with ALA

Figure 4 demonstrates clear photodynamic killing effects on subsets of PBMCs with ALA. In the subpopulations of T cells (CD3^+^, CD4^+^, and CD8^+^) much smaller killing effects than those in the CD3^−^CD19^+^ B cells, CD3^−^CD56^+^ NK cells, and CD11c^+^CD14^+^ dendritic cells were seen.

There are big variations of the PDT killing effects on the subpopulations of T cells among various samples (Appendix A).

### 3.4. Effects of ALA-PDT on Cytokines of PBMCs

Many cytokines of PBMCs were affected after ALA-induced PDT. As shown in Figure 5, the data are presented from high to low levels of log decrease.

The log decrease is a log_2_(ALAi − PDTi), where ALAi is the cytokine fluorescence intensity in the control group with ALA alone and PDTi is the cytokine fluorescence intensity in the PDT group with ALA plus light. ALA-PDT led to various effects on different cytokines. For example, ALA-PDT clearly reduced the amount of MIP-1 alpha, whereas little effect was seen on the IL-12(p70). There are variations of the PDT effects on the levels of cytokines, particularly in the cases of IL-6, IFN-gamma, and IL-1ra (Appendix A).

### 3.5. Effects of ALA-PDT on Exosomes of PBMCs

ALA-PDT also affected various types of exosomes from PBMCs. Figure 6 presents the data from high to low levels of log decrease. The log decrease is a log_2_(ALAi − PDTi), where ALAi is the fluorescence intensity of an exosome surface marker in the control group with ALA alone and PDTi is the fluorescence intensity of an exosome surface marker in the PDT group with ALA plus light. The higher log decrease value is, the lower level of an exosome is. ALA-PDT changed the amounts of different exosomes of PBMCs. For example, ALA-PDT decreased more the amount of CD29 exosomes than CD133-1 exosomes. There are relatively big variations of the PDT effects on the amounts of exosomes, particularly in the exosomes with the surface markers of HLA-ABC, CD69, and CD133-1 (Appendix A).

### 3.6. Effects of Light Alone on Subsets, Exosomes, and Cytokines of PBMCs

Light alone did not affect the cell survivals of subsets of PBMCs. No effects of light alone on the cytokine levels were seen, except for a minor decrease in IFNγ. The effects on the exosomes were also not found after the treatment with light alone, except a slightly stimulatory response of the CD133-1.

### 3.7. Electron Microscopy of Exosomes

Typical images of exosomes made by electron microscopy are shown in the Figure 7. The size of the exosomes in this study was confirmed to be around 85 nm.

Exosomes appeared to have a spherical shape with a typical central inflation caused by samples preparation and drying. These morphological data clearly demonstrate the exosomal identity of samples isolated using a combination of flow cytometry and magnetic beads.

## 4. Discussion

PDT with ALA is a clinically established modality for the treatment of malignant and premalignant diseases [1,16]. Although numerous reports have confirmed the mechanism of PDT action involved in the killing of the diseased cells via necrosis and apoptosis [17,18,19], no information so far is available for the effects of ALA-PDT on the exosomes of human PBMCs. To understand and explore the potential of exosomes has recently become an urgent issue for the improvement of the clinical ALA-PDT efficacy.

Generally, there are variations of ALA-induced PpIX production and PDT killing effects in the samples from different donors (Figure 2 and Figure 4). The reasons for such individual variations are not known, although we have noticed a similar situation in our other studies [20]. This might be due to the fact that the PBMC samples were taken from various donors with different genders and ages. Light alone did not affect the survivals of all subsets of PBMCs. ALA alone caused no cytotoxicity of PBMCs either, except CD3^−^CD56^+^ NK cells (Figure 3). Since NK cells generally appear to be more fragile to any treatments, the reason for the ALA dark toxicity on the NK cells could be due to a relatively long cell culture in vitro with ALA for a total of 52 h. The CD8^+^ T cells with a low ALA-PpIX production were killed much less than other subsets of PBMCs after light exposure; while the CD3^−^CD19^+^ B cells were killed more with a high ALA-PpIX production after light irradiation (Figure 2 and Figure 4).

In physiological and pathological conditions, almost all types of cells release exosomes. Such exosomes contain biological and genetic molecules and carry them to other cells for cell communication and epigenetic regulation. Several techniques have been established to isolate exosomes. They include ultracentrifugation techniques (sequential ultracentrifugation and gradient ultracentrifugation), size-based isolation (ultrafiltration and size-exclusion chromatography), polymer precipitation, etc. These techniques are still under optimalization. In the present study, we employed an immunoaffinity capture technique with specific binding between exosome markers and immobilized antibodies (ligands) measured with flow cytometry. The technique has several advantages over other methods including easy use, separation of exosomes from different origins, and high purity of exosomes with no chemical contamination [21]. In addition, we used electron microscopy to morphologically confirm the exosomes measured with flow cytometry (Figure 7).

ALA-PDT reduced the levels of most cytokines studied (Figure 5), although with different degrees of such reduction in various cytokines. Since almost all cytokines included in this study are pro-inflammatory, except IL-1ra that is anti-inflammatory (Table 1), these consistent results may suggest that ALA-PDT of PBMCs results in a strong anti-inflammatory effect.

This may also be true for the level of IFN-gamma, a well-known cytokine for pro-autoimmune diseases, which was clearly reduced after ALA-PDT (Table 1). This finding makes one tempting to speculate if ALA-PDT might have the potential for reducing the cytokine storm in COVID-19 cases [22].

ALA-PDT also decreased the amounts of all exosomes of PBMCs (Figure 6, Table 2). It is not clear if such decreased amounts of exosomes were due to a direct destruction of PDT effect, PDT-mediated damage to those parent cells making them unable to produce exosomes, or both.

However, the amount of CD4^+^ exosomes is higher than CD8^+^ exosomes after ALA-PDT (Table 2), even though CD4^+^ cells produced more ALA-induced PpIX and killed more than CD8^+^ cells. The reason for is not known, but our previous studies have confirmed that ALA-PDT could induce apoptosis of leukemia and lymphoma with the formation of apoptotic bodies [17,18,19]

It is interesting to note that both CD40 and CD19 as surface markers for B cells were apparently decreased, probably because the CD3^−^CD19^+^ B cells produced a high amount of PpIX from ALA (Figure 2) and were killed more after light exposure (Figure 4). A reduced amount of CD69 exosomes was also seen after PDT treatment (Table 2). Such exosomes are believed to activate and proliferate lymphocytes [23].

CD25 is an activating marker for lymphocytes and macrophage, particularly for T cells. Interestingly, the amount of CD25 exosomes was not affected very much as expected by ALA-PDT probably due to few activated T cells in the PBMC samples from healthy donors.

HLA-DRDPDQ is the surface marker for antigen-presenting cells (APC). Since APC is heavily involved in presenting the MHC class II molecules to CD4^+^ T cells for the rejection of organ transplantation [24,25,26,27,28], the decrease in the amount of the APC-derived exosomes may indicate that such a reduction of HLA-DRDPDQ exosomes by ALA-PDT may have an impact on reducing such organ rejection. Similarly, a reduced amount of HLA-ABC exosomes, which are involved in presenting the MHC class I molecules to CD8^+^ T cells, was seen. Such a finding suggests that ALA-PDT of PBMCs may reduce the MHC class I molecule-mediated cytotoxic CD8^+^ immunity.

Both CD133-1 [29] and SSEA-4 [30] are markers for stem cells with lower reduction of exosomes after PDT. This could be due to the fact that stem cells normally are resting with a low energy demand, so that the rate of heme synthesis is minimal, through which a low amount of ALA-PpIX is produced. As a result, PDT with ALA has a minimal destructive effect on these cells.

CD9 and CD41b exosomes are involved in platelet activation and aggregation and were also used as markers for rejection of heart transplantation [31]. After ALA-PDT, the amount of the exosomes was reduced, suggesting a favorable response to this treatment.

Cell adhesion molecule are cell surface proteins. They help cells to bind other cells to maintain cells/tissue functional. They also play important roles in cell growth and death, contact inhibition, etc. ALA-PDT also decreased several exosomes that are involved in cell adhesion, including CD29, CD9, CD31, CD41b, CD44, and CD81 exosomes. The biological implications of such a PDT effect are not known and warrant further investigation.

## 5. Conclusions

This is the first report to study the effects of ALA-PDT on cytokines and exosomes of human healthy PBMCs. ALA-PDT reduced all cytokines and exosomes studied, although there was variation among individual PBMC samples. This reduction is consistent with ALA-PDT-mediated survivals of subsets of PBMCs. More specifically, the ALA-PDT treatment apparently decreased all pro-inflammatory cytokines included, suggesting that this treatment provides a strong anti-inflammatory effect. In addition, the treatment has decreased the levels of different types of exosomes with a particular interest in the HLA-DRDPDQ exosome, which plays an important role in presenting the MHC class II molecules to CD4^+^ T cells for rejection of organ transplantation and autoimmune diseases [24,25,26,27,28].

## Figures and Tables

**Figure 1 biomedicines-10-00232-f001:**
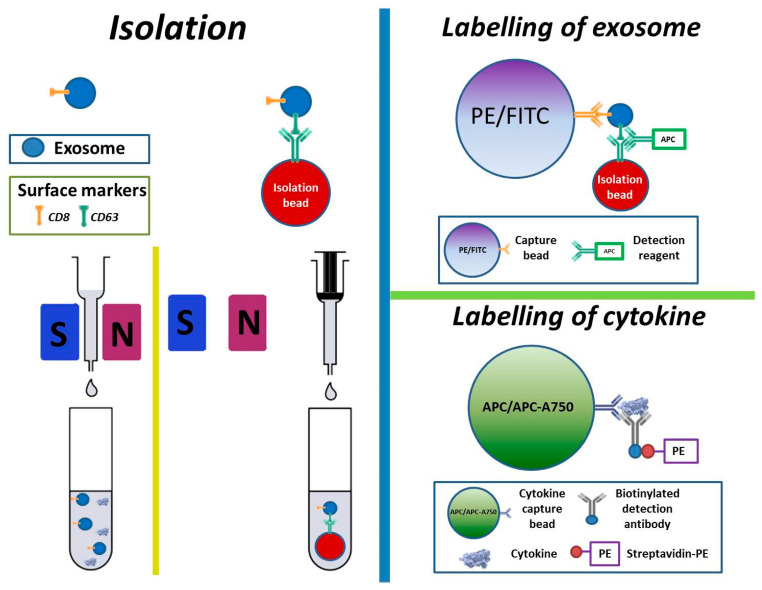
A simplified graphic description of the procedure for isolation and labelling of exosomes. FITC = Fluorescein isothiocyanate, APC = allophycocyanin, PE = phycoerythrin, S = south (on magnet), N = north (on magnet).

**Figure 2 biomedicines-10-00232-f002:**
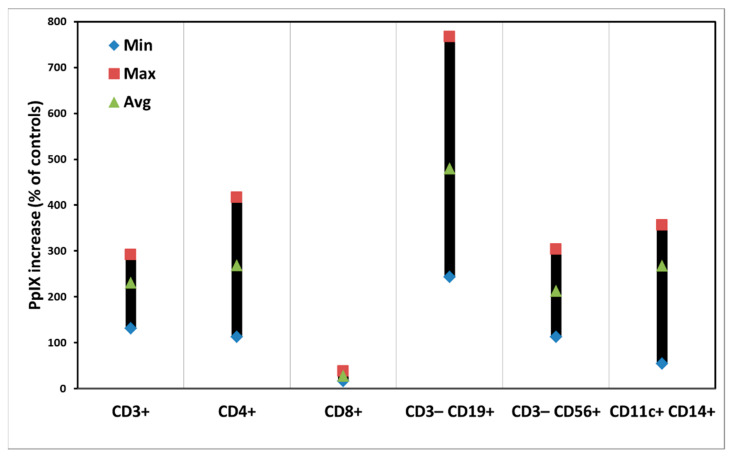
Intracellular amounts of PpIX in individual subsets of PBMCs after ALA incubation for 4 h. The data are presented as minimal, maximal, and average values.

**Figure 3 biomedicines-10-00232-f003:**
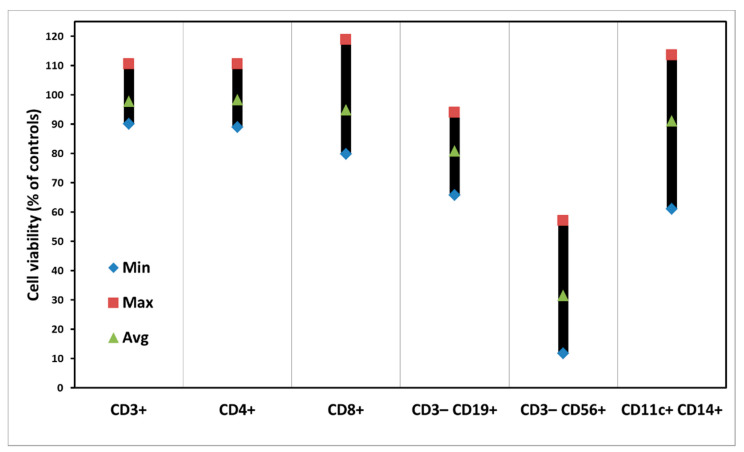
Dark toxicity of cells in individual subsets of PBMCs after ALA incubation for 52 h. The data are presented as minimal, maximal, and average values.

**Figure 4 biomedicines-10-00232-f004:**
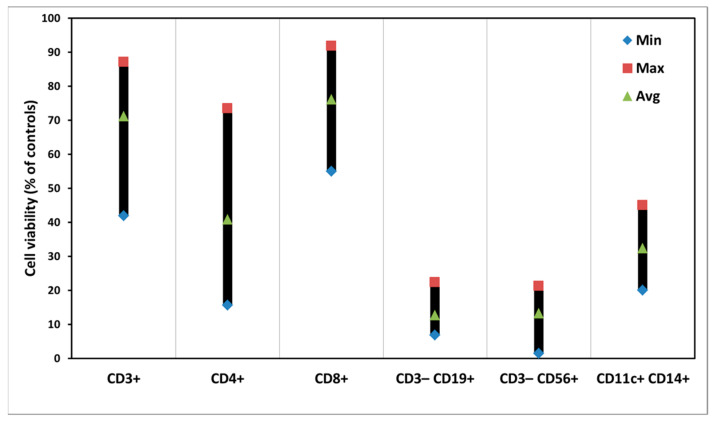
Photodynamic effects on cells in individual subsets of PBMC with ALA. The cells were incubated with ALA at a concentration of 3 mM for 4 h, followed by light exposure at a dose of 180 J/cm^2^. The cells then continued to be cultured for another 48 h before the measurements of cell viabilities. The data are presented as minimal, maximal, and average values.

**Figure 5 biomedicines-10-00232-f005:**
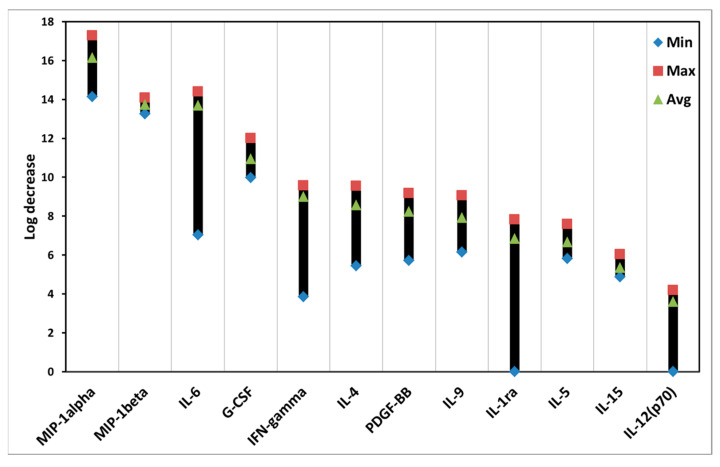
Photodynamic effects on cytokines of PBMCs with ALA. The PDT treatment was the same as that in Figure 4. The data are presented as minimal, maximal, and average log decrease values. The log decrease is a log_2_(ALA_i_ − PDT_i_), where ALA_i_ is a cytokine fluorescence intensity in the control group with ALA alone and PDT_i_ is a cytokine fluorescence intensity in the PDT group with ALA plus light. The higher log decrease value, the lower level of a cytokine.

**Figure 6 biomedicines-10-00232-f006:**
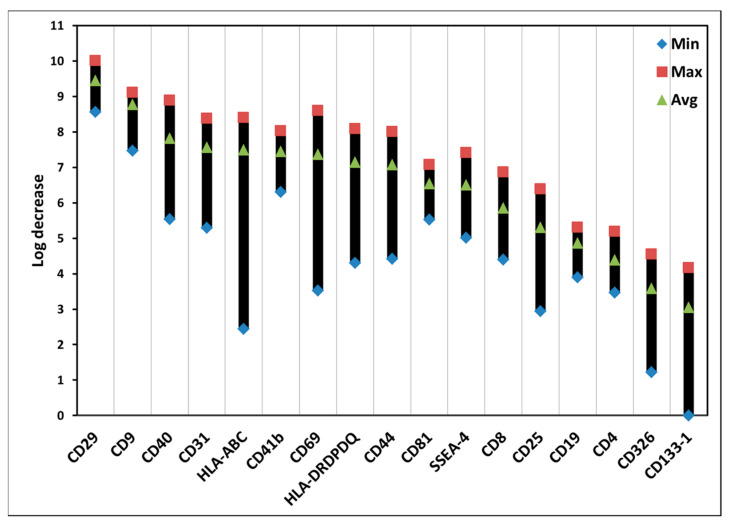
Photodynamic effects on exosomes of PBMCs with ALA. The PDT treatment was the same as that in Figure 4. The data are presented as minimal, maximal, and average log decrease values. The log decrease is a log_2_(ALAi − PDTi), where ALAi is a surface marker fluorescence intensity of an exosome in the control group with ALA alone and PDTi is a surface marker fluorescence intensity of an exosome in the PDT group with ALA plus light. The higher log decrease value, the lower level of a surface marker fluorescence intensity of an exosome.

**Figure 7 biomedicines-10-00232-f007:**
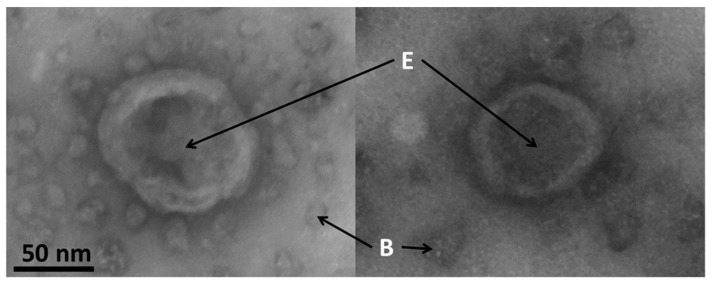
Typical images of exosomes made by electron microscopy. E: exosomes; B: magnetic beads. Detailed information on the preparation of the samples is found in Section 2.8. Scale bar = 50 nm.

**Table 1 biomedicines-10-00232-t001:** Effects of ALA-PDT on cytokines ^1^.

Cytokine	PDT Effect (Avg.)	Cell Type	Possible Biological Function
MIP-1alpha (macrophage inflammatory protein-1 alpha)	16.1	Macrophage	Pro-inflammation
MIP-1beta (macrophage inflammatory protein-1beta)	13.7	Macrophage	Pro-inflammation
IL-6 (interleukin-6)	13.7	Macrophage	Pro-inflammation
G-CSF (granulocyte colony stimulating factor)	10.9	Macrophage and other cells	Growth stimulation of white blood cells
IFN-gamma (interferon gamma)	9	T-cell and NK cell	Pro-inflammation and worsen autoimmune diseases
IL-4 (interleukin-4)	8.6	Mast cell, T-cell, granulocyte	Pro-inflammation
PDGF-BB (platelet-derived growth factor-BB)	8.2	Platelet, macrophage, and other cells	Wound healing and repair blood vessel
IL-9 (interleukin-9)	7.9	CD4^+^ T cell	Pro-inflammation
IL-1ra (interleukin-1 receptor antagonist)	6.8	Macrophage and other cells	Anti-inflammation
IL-5 (interleukin-5)	6.7	T cell, granulocyte, and other cells	Pro-inflammation
IL-15 (interleukin-15)	5.4	Macrophage	Pro-inflammation
IL-12(p70) (interleukin 12p70)	3.6	Macrophage	Pro-inflammation

^1^ Note: Since the types of cells and possible biological functions of the cytokines studied are complicated with different types of cells and multi-functions, the table only lists main cell types and functions of the cytokines.

**Table 2 biomedicines-10-00232-t002:** Effects of ALA-PDT on exosomes ^1^.

Surface Marker	PDT Effect (Avg.)	Cell Type	Possible Biological Function
CD29	9.4	White blood cells	Cell adhesion
CD9	8.8	Lymphocyte, macrophage	Platelet activation and aggregation and cell adhesion and migration
CD40	7.8	B-cell, macrophage	Cell proliferation and signal transduction
CD31	7.6	White blood cells	Cell adhesion, activation, and migration
HLA-ABC	7.5	Nucleated cells	MHC class I molecules presented to CD8^+^ T cells
CD41b	7.4	Stem cell, platelet	Cell adhesion and platelet aggregation
CD69	7.4	White blood cells	Lymphocyte activation and proliferation
HLA-DRDPDQ	7.1	Antigen-presenting cell	MHC class II molecules presented to CD4^+^ T cells
CD44	7.1	White blood cells	Cell adhesion
CD81	6.5	White blood cells but granulocyte	Cell adhesion
SSEA-4	6.5	Embryonic stem cell	Pluripotent stem cell marker
CD8	5.9	T cell	Cytotoxic T cell marker
CD25	5.3	Lymphocyte and macrophage	Lymphocyte activation
CD19	4.9	B cell	B cell marker
CD4	4.4	T cell	Helper T cell marker
CD326	3.6	T cell, dendritic cell, epithelial cell	Epithelial cell marker unknown functions on immune cells
CD133-1	3.1	Stem cell and endothelial cell	Stem cell marker

^1^ Note: Since the cell types and possible biological functions of the exosomes studied are complicated with multi-cell types and multi-functions, the table only lists main cell types and functions of the exosomes.

## Data Availability

The data presented in this study are available in the Appendix A.

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
