# Peer review of "Photodynamic Effects with 5-Aminolevulinic Acid on Cytokines and Exosomes in Human Peripheral Blood Mononuclear Cells"

_biomedicines, 2022, doi:10.3390/biomedicines10020232_

Round 1

Reviewer 1 Report

The manuscript contains the study of PDT effect on exosomes and cytokines.

There are some mistakes as typos which have to be corrected.

  1. line1: exosomeS
  2. line 104: 2x106 CFU/ mL
  3. line 112: add time or light dose
  4. Please add the used light doses for clinical ALA-PDT
  5. A short discussion of the obtained results on the overall PDT efficiency of ALA can be added. 

Author Response

  1.  “line1: exosomeS.” the S has been changed with a lower case.
  2. “line 104: 2x106 CFU/ mL” We used 2 million cells per mL, the sentence has been changed to 2x106 cells/mL, as we did not use CFU. 
  3. “line 112: add time or light dose.” This has been added in the revised manuscript, and also moved from the section 2.5 to 2.4.
  4. “Please add the used light doses for clinical ALA-PDT.” The light doses of clinical ALA-PDT depend on the medical indications with huge variations in terms of size/depth of lesions and on the light wavelengths used. It also depends on the nature of a lesion whether it is a solid tissue or hematological cell suspension sample. There is no established ALA-PDT of a hematological disease yet, although we have our ongoing clinical trials of ALA-PDT of PBMCs in the patients with graft versus host disease [1] (Ref. 14 in the MS).
  5. “A short discussion of the obtained results on the overall PDT efficiency of ALA can be added.” This has been discussed in the 2nd paragraph of the revised MS.

Reference

1. Christensen E, Foss OA, Quist-Paulsen P, Staur I, Pettersen F, Holien T, et al. Application of Photodynamic Therapy with 5-Aminolevulinic Acid to Extracorporeal Photopheresis in the Treatment of Patients with Chronic Graft-versus-Host Disease: A First-in-Human Study. Pharmaceutics. 2021;13(10). 10.3390/pharmaceutics13101558 

Reviewer 2 Report

The manuscript represents a study of photodynamic effect with protoporphyrin IX (PpIX) from 5-aminolevulinic acid (ALA) on exosomes and cytokines in human healthy peripheral blood mononuclear cells (PBMCs). Such a study has not been conducted before and can therefore contribute to PDT research and improve clinical applications with its findings.

The manuscript is well written, nicely organized and easy to read, the experiments are described in detail, and the results are clearly presented and explained, with appropriate discussion and conclusions. References are appropriate, but I think that in the introductory part some more examples of PDT effect on exosomes available in the literature could be given.

My main concern about this work is small number of samples (4 donors), especially because there are big variations of the PPIX production in PBMCs, dark toxicity with ALA and the PDT killing effects among samples. Maybe that could be discussed at least a little more. Furthermore, only one concentration of ALA was tested. It should be commented on how the ALA concentration and incubation time were decided.

The only objection I have about the work is that I don’t see the “light only” control in the experiments. A fairly high fluence rate (100 mW/cm2) was used and a total light dose is quite large (180 J/cm2), so the effect of light alone on PBMCs and on cytokine and exosome levels should definitely be checked.

Author Response

“..but I think that in the introductory part some more examples of PDT effect on exosomes available in the literature could be given.” We have added sentences as to the PDT effect on exosomes with relevant references in the Introduction (3rd paragraph) of the revised MS.

“My main concern about this work is small number of samples (4 donors), especially because there are big variations of the PPIX production in PBMCs, dark toxicity with ALA and the PDT killing effects among samples. Maybe that could be discussed at least a little more.” We have followed the comment and discussed it in the 2nd paragraph of the revised MS.

“Furthermore, only one concentration of ALA was tested. It should be commented on how the ALA concentration and incubation time were decided.” The concentration of 3 mM ALA dose was chosen based on our previous study [1] (Ref. 15 in the revised MS). This information has been added in the 2.3 section of the revised MS. 

“The only objection I have about the work is that I don’t see the “light only” control in the experiments. A fairly high fluence rate (100 mW/cm2) was used and a total light dose is quite large (180 J/cm2), so the effect of light alone on PBMCs and on cytokine and exosome levels should definitely be checked.” We appreciate the relevant comment and have added a new section of 3.6 to present our results of the effects of light alone on subsets, exosomes and cytokines of PBMCs in the revised MS.

Reference

1. Darvekar S, Juzenas P, Oksvold M, Kleinauskas A, Holien T, Christensen E, et al. Selective Killing of Activated T Cells by 5-Aminolevulinic Acid Mediated Photodynamic Effect: Potential Improvement of Extracorporeal Photopheresis. Cancers (Basel). 2020;12(2):377. 10.3390/cancers12020377

Round 2

Reviewer 2 Report

The manuscript has been improved. Minor flaws have been removed, all questions have been answered and I no longer have any reservations that it is worth publishing.